# Effects of UV Radiation on the Carbonation of Cement-Based Materials with Supplementary Cementitious Materials

**Haoyuan Li [1,2], Zhonghe Shui [1,2], Ziyan Wang [1,2,3,\*] and Xunguang Xiao [4]**

1 School of Materials Science and Engineering, Wuhan University of Technology, Wuhan 430070, China; lihaoyuan@whut.edu.cn (H.L.); zhshui@whut.edu.cn (Z.S.)
2 State Key Laboratory of Silicate Materials for Architectures, Wuhan University of Technology, Wuhan 430070, China
3 Guizhou Construction Science Research and Design Institute Limited Company of CSCEC, Guiyang 550004, China
4 Advanced Engineering Technology Research Institute of Zhongshan City, Wuhan University of Technology, Zhongshan 528400, China; whutxxg1992@163.com
\* Correspondence: zy_w@whut.edu.cn

**Abstract:** Solar light with high-energy ultraviolet (UV) radiation acting on the surface of cement-based materials easily changes the properties of cement-based materials by affecting their carbonation reaction. In order to elucidate the difference in the carbonation process under UV radiation in cement-based materials with different supplementary cementitious materials (SCMs), the carbonation depth (apparent pH values), chemical composition (XRD, FTIR, and TG analysis), and mechanical properties (compressive strength and microhardness) of cement-based materials were evaluated. The results revealed that UV radiation acting on the surface of cement-based materials accelerated the carbonation reaction, which enhanced the decrease rate of pH and formation of stable calcite, thereby improving the macromechanical and micromechanical properties of cement-based materials. In addition, the carbonation process under UV radiation differs according to the added SCM. In particular, silica fume substantially increased the carbonation of cement-based materials under UV radiation, resulting in a 53.3% increase in calcium carbonate coverage, a 10.0% increase in compressive strength, and a 20.9% increase in mean microhardness, whereas the incorporation of blast furnace slag resulted in a smaller effect on UV irradiation-induced carbonation. In addition, UV radiation facilitates the crystallographic transformation process of cement-based materials containing metakaolin, resulting in more stable crystals of carbonation products. This study provides a theoretical framework and serves as an important reference for the design of cement-based materials under strong UV radiation for practical engineering applications.

**Keywords:** UV radiation; supplementary cementitious materials; cement-based materials; carbonation

## 1. Introduction

In regions with longer average daily light hours and higher solar radiation intensity, such as the southeast coastal areas and western plateau of China, the surface properties of cement-based materials are more susceptible to variations when exposed to solar light [1]. In addition to the effects due to the temperature changes under infrared radiation, the effects of ultraviolet (UV) radiation, which has higher radiant energy, on the properties of cement-based materials cannot be ignored [2,3]. Among the solar wavelengths, UV wavelengths account for only 5%–7%; however, they carry high levels of radiation that cause degradation, aging, and curing of materials [4]. In addition, UV radiation can cause chemical reactions on the surface of cement-based materials, thereby affecting the composition of their product phases and mechanical properties [5,6]. Shui et al. found that UV radiation can increase the quantity of calcite formed on the surface of the cement paste and change the physical properties of the surface [5]. Lan et al. noted the promoted carbonation of

concrete under UV radiation by comparing various concrete curing processes [6]. Although previous studies have clarified the effect of UV radiation on the carbonation behaviour of cement-based materials, the precise mechanism of action of UV radiation still needs to be established [7].

Concrete carbonation is a chemical reaction that occurs between $CO_2$ and cement hydration products. Due to the porous structure of concrete, $CO_2$ from the environment diffuses through the gas phase of the concrete pores to the interior of the concrete and dissolves in the pore water, thereby consuming alkaline substances, such as calcium hydroxide and lowering the pH of the pore solution [8–11]. The reduction in the alkalinity of concrete causes the failure of the highly alkaline passivation film of the reinforcement, resulting in corrosion; thus, this is considered the main cause of the deterioration in the durability of reinforced concrete [12,13]. Furthermore, the difference in the molar volume of the reactants and products of the carbonation reaction significantly affects the microstructure of the cement-based materials. The carbonation reaction fills the capillary pores of the concrete with carbonation products, thereby increasing the density and strength of the concrete [14,15].

In modern cement-based materials, supplementary cementitious materials (SCMs) are added to achieve economic and environmental benefits. SCMs affect the composition of the hydration products, resulting in different carbonation processes of cement-based materials under solar exposure [16,17]. Therefore, further investigation of the effect of UV radiation on the carbonation process of SCMs and the corresponding reaction mechanism is important for the effective use and maintenance of cement-based structures [18].

This study aimed to systematically investigate the effects of UV radiation on the carbonation behaviour of cement-based materials and clarify the related mechanism and thus establish a theoretical and experimental basis for engineering cement-based materials. Three SCMs, namely silica fume (SF), blast furnace slag (GGBS), and metakaolin (MK), were selected to partially replace cement in the preparation of cement pastes, which were subsequently subjected to UV and standard (non-UV) curing after hardening [19–23]. The pH of the paste at different depths from the exposed surface was evaluated for each layer to characterise the effect of the UV radiation depth. Furthermore, to investigate the effects of UV radiation on the evolution of the carbonation behaviour and mechanical properties of different cement-based materials, X-ray diffraction (XRD), Fourier transform infrared (FTIR) spectroscopy, thermogravimetric (TG) analysis, compressive strength test, and Vickers hardness test were conducted to investigate their phase composition, compressive strength, and microhardness. These analyses can aid the formulation of important guidelines for the design and durability prediction of cement-based materials exposed to strong UV radiation.

## 2. Experimental

### 2.1. Raw Materials and Mixing Ratios

Ordinary Portland cement (OPC) and three types of SCMs, viz., SF, GGBS, and MK were used in this study. The chemical compositions of these raw materials are summarised in Table 1. Their particle size distributions are shown in Figure 1, and the results of the FTIR analysis are presented in Figure 2. The apparent densities of OPC, SF, GGBS, and MK are 3.11, 2.21, 2.82, and 2.49g/cm$^3$, respectively.

**Table 1.** Chemical compositions of the raw materials (wt%) used in this study.

| Raw Materials | Chemical Composition | | | | | | | | |
|---|---|---|---|---|---|---|---|---|---|
| | $SiO_2$ | $CaO$ | $Al_2O_3$ | $Fe_2O_3$ | $SO_3$ | $MgO$ | $Na_2O$ | $K_2O$ | Others |
| OPC | 25.15 | 54.65 | 7.42 | 3.07 | 2.68 | 1.53 | 0.18 | 0.98 | 4.33 |
| SF | 96.50 | 0.05 | 0.40 | 0.00 | 0.46 | 0.11 | 0.04 | 0.08 | 2.37 |
| GGBS | 32.31 | 42.66 | 12.61 | 0.32 | 2.50 | 6.88 | 0.49 | 0.42 | 1.80 |
| MK | 48.82 | 0.83 | 42.27 | 0.00 | 0.15 | 0.26 | 0.07 | 0.05 | 7.56 |

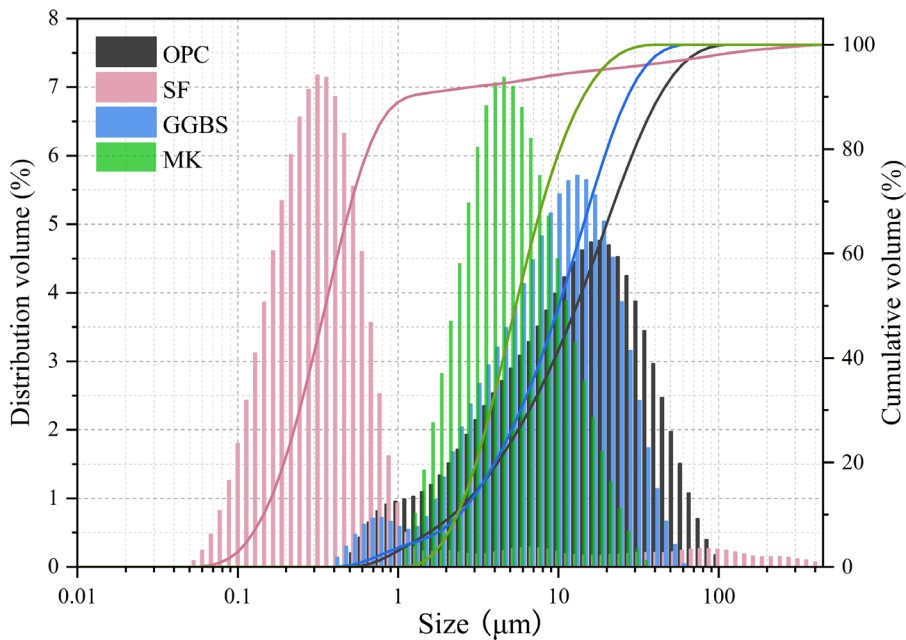

**Figure 1.** Particle size distribution of the raw materials used in this study.

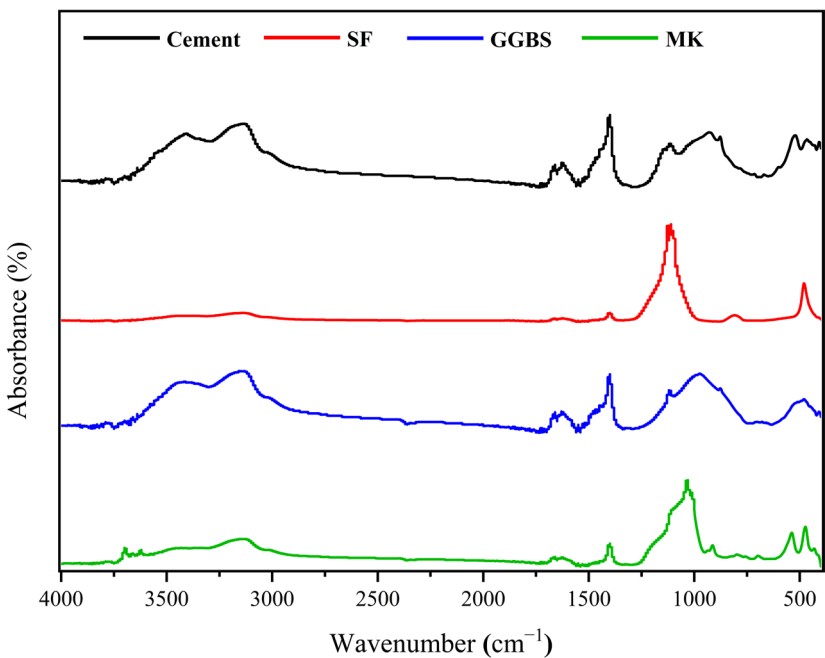

**Figure 2.** Infrared spectral analysis of the raw materials used in this study.

In order to investigate the change pattern and reaction mechanism of cement-based materials under UV radiation, the pastes were selected as the study specimens. As shown in Table 2, in all the cement mixtures, the water-binder (w/b) ratio was kept constant at 0.45. In addition to preparing the pristine cement paste, mixtures of cement with SF, GGBS, and MK at 10% of the cement by mass were prepared. The cement pastes were mixed well and poured into 20 mm × 20 mm × 20 mm iron moulds. Subsequently, they were placed in a standard curing chamber, and their surfaces were sealed with a plastic film to prevent moisture evaporation. After 1 d of curing, the specimens were demoulded and transferred to a UV-weathering chamber for the ageing test.

**Table 2.** Compositions of pastes.

| Sample Code | Cementitious Materials (wt%) | | | | w/b Ratio |
|---|---|---|---|---|---|
| | OPC | SF | GGBS | MK | |
| C | 100 | - | - | - | |
| C-SF | 90 | 10 | - | - | |
| C-GGBS | 90 | - | 10 | - | 0.45 |
| C-MK | 90 | - | - | 10 | |

### 2.2. UV-Irradiation Test

The UV-curing conditions were implemented using a Shanghai Yiheng UV-weathering chamber a UV-weathering chamber (Yiheng, China) equipped with a UVA lamp (American ATLAS) UVA lamp (ATLAS, Mount Prospect, IL, USA), as shown in Figure 3a. In order to simulate the specific natural environment with high temperature and high UV radiation, the experimental program was set as ① daylight mode: temperature of 60 °C, irradiance at 1.1 W/cm$^2$/nm, single cycle time of 4 h; ② night condensation mode: temperature of 40 °C, irradiance at 0 W/cm$^2$/nm, single cycle time of 2 h [24], as shown in Figure 3b,c. These two modes formed a day/night cycle during which the $CO_2$ concentration in the chamber was maintained at 0.04%, which is consistent with the outer atmosphere; no additional $CO_2$ was introduced into the chamber.

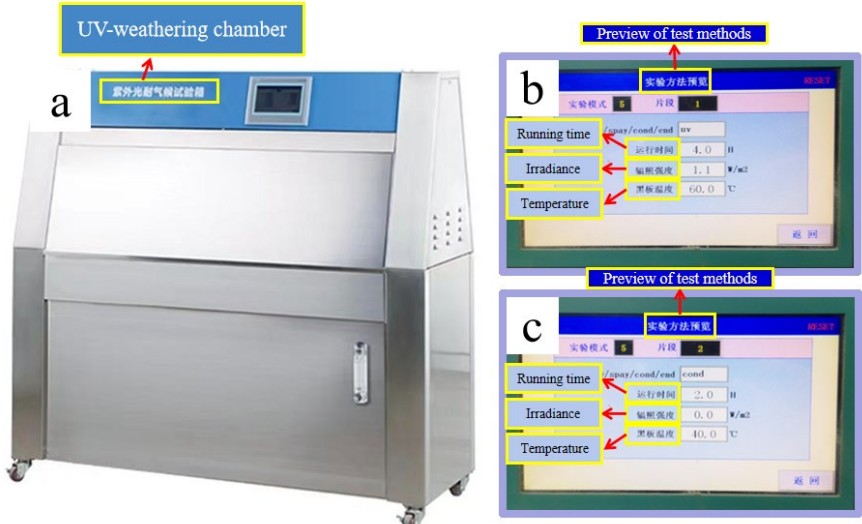

**Figure 3.** (**a**) UV-weathering chamber; (**b**) testing program in mode 1; (**c**) testing program in mode 2.

To exclude interference from other environmental factors and to compare the effects of UV radiation, reference group (denoted as the UV group) samples were set up and placed inside the test chamber along with the experimental group (denoted as the NUV group). As shown in Figure 4, the UV group and the NUV group were placed in different areas of the same chamber to maintain the same temperature, humidity, and $CO_2$ concentration conditions. The difference was that the UV group was placed inside a metal plate and exposed to direct UV radiation during daylight mode, whereas the NUV group was placed on the opposite side of the metal plate and was never illuminated with UV light.

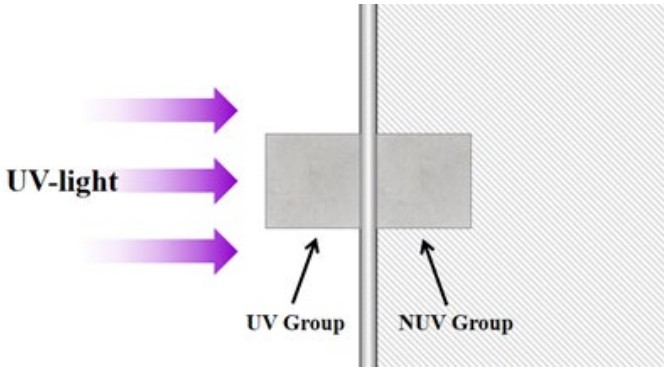

**Figure 4.** Schematic of specimen placement in the test chamber.

*2.3. Material Characterisation*

In this study, powder samples used for compositional analysis were obtained from different depths of the specimens, and the hydration of the samples was terminated using the solvent exchange method with alcohol. In addition, specimens for microhardness tests were ground and polished after resin impregnation; successively finer abrasives starting from 400, 800, and 1200 grit papers to 3 μm diamond suspension were used for grinding and polishing.

The pH of the samples was measured using a PHS-3E pH meter (Shanghai Yoke Instrument Co., Ltd., Youke, China). Initially, the powdered samples were dissolved in distilled water at a mass ratio of 1:100, and the solution was ultrasonicated for 10 min to accelerate ion extraction. After an additional 10 min of precipitation, the clear upper liquid was retrieved for pH measurement.

XRD and attenuated total reflectance FTIR spectroscopy was conducted to qualitatively characterise the compositions of the pastes. XRD was carried out on a Malvern Panalytical Empyrean diffractometer (Malvern Panalytical, Westborough, MA, USA) (Cu $K_\alpha$, $\lambda$ = 1.54 Å); the samples were scanned between 5 and 65° in *2θ* with a step size of 0.2°/s. FTIR spectra were recorded using a Nicolet Is10 spectrometer (Thermo Fisher Scientific, Waltham, MA, USA) in the frequency range of 400–4000 cm$^{-1}$.

TG analysis was conducted using a Netzsch STA 449 F5 analyser (Netzsch, Selb, Germany) to quantify the contents of hydrated and carbonated phases. The samples were heated in the temperature range of 30–1000 °C at a heating rate of 20 °C/min under a constant nitrogen flow of 20 mL/min.

Furthermore, the mechanical properties of the pastes were evaluated by measuring their compressive strength and surface Vickers hardness. Compressive strength tests were performed on three specimens from each group, and the averaged results were reported. The tests were conducted using an ETM305D universal testing machine (Wance, Shenzhen, China) at a loading rate of 100 N/s, as shown in Figure 5. The Vickers hardness of a material can reveal its local mechanical properties based on its composition and microstructure. For microhardness evaluation, 64 surface points were selected uniformly on the test specimen, and the measurements were performed using a TMVP-1 Microhardness tester (Beijing Times Peak Technology Co, Beijing, China).

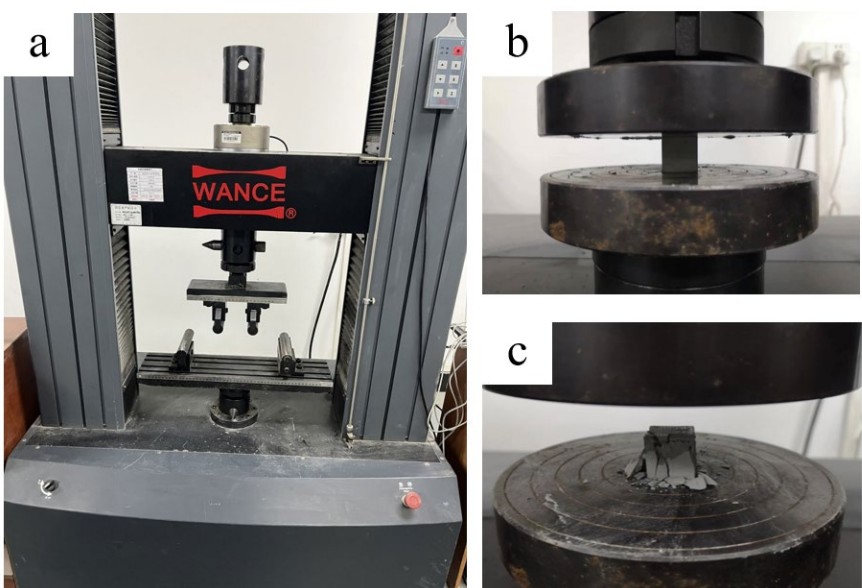

**Figure 5.** (**a**) ETM305D universal testing machine; (**b**) testing; (**c**) broken diagram.

## 3. Results and Discussion

### 3.1. Change in the pH of Cement-Based Materials during Their Curing

The apparent pH values of the samples retrieved from different depths of the four different pastes (C, C-SF, C-GGBS, and C-MK; see Table 2) were measured after curing with or without UV radiation for 7, 14, 28, and 56 d, using the method described in Section 2.3, and the pH variation trend is shown in Figure 6. The apparent pH can effectively reflect the change in the degree of carbonation because the carbonation reaction consumes portlandite and other alkaline hydration products, causing a decrease in the alkalinity of the pore solution [25,26].

As shown in Figure 6, remarkable differences were observed in the pH values between the UV and NUV samples; the pH values of all the samples increased with the increase in depth from the surface and eventually became constant. The interior of the pastes still presented a high level of alkalinity, which is associated with a decline in the reaction rate and degree of carbonation in the interior. This is because of the reduction in the contact area between the interior of the specimens and $CO_2$ in the ambient atmosphere [27]. The pH variation data in Figure 6a–d indicate that the pH variation trends of the four samples, viz., C, C-SF, C-GGBS, and C-MK, were similar at 7 days. The pH values of the NUV groups that were not exposed to UV light decreased in the depth range of approximately 0.25–0.5 mm, whereas the depth affected by the carbonation reaction increased to ~1 mm for the UV groups. As the curing time increased, the pH value decreased further, and the depth and degree to which the sample was affected by the carbonation reaction increased significantly. The pH change trend at 14 days was similar to that at 7 days [28]. Compared with the apparent pH of the interior part of the specimen, which was not involved in the carbonation reaction, the depth of pH change in the UV and NUV groups reached 0.5–0.75 and 1.5 mm, respectively. As the curing time increased further, a more notable difference appeared between different pastes, as shown in Figure 6c. This is attributed to the different hydration characteristics and microstructures of the pastes according to the added SCM, which induces a distinctive promotional effect of UV radiation on the carbonation reaction. Upon curing for 28 days, the depth of change in the NUV samples reached 1.5–2 mm. On the other hand, the depth of change in group C, C-SF, and C-GGBS under UV radiation increased to approximately 1.5, 2.5, and 3 mm, respectively, indicating that after a longer UV-curing time, a continuous promotional effect of the UV radiation on the carbonation of cement-based materials mixed with SF and GGBS occurred, whereas the depth of change in the pristine cement slurry under UV exposure did not continue to increase. In addition, the

pH values of the C-MK samples no longer exhibited significant differences after 28 days, either with or without UV radiation, probably because of the strong Pozzolanic and physical filling effects of MK, which increased the hydration level and density of the matrix [29]. Thus, the optimisation of the pore structure in C-MK hinders the propagation of $CO_2$ and the diffuse reflection of UV light, inhibiting further carbonation [30].

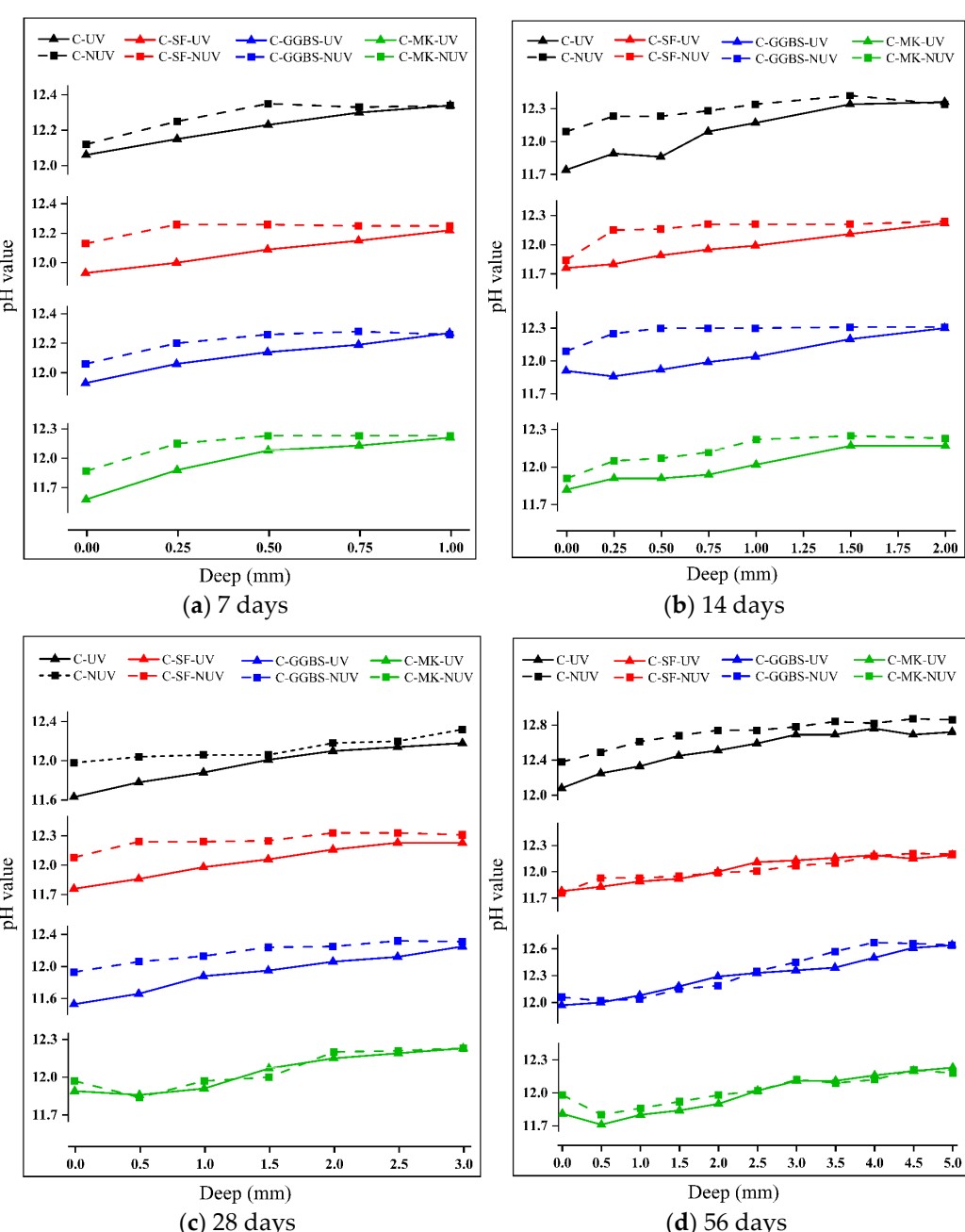

**Figure 6.** Evolution of the pH at different depths of the samples with the curing time.

When the curing time was increased further to 56 days, as shown in Figure 6d, none of the mixed samples showed significant differences, and only the pristine cement partially retained the promoting effect of UV radiation. These results indicate that UV radiation accelerates the carbonation process but does not significantly increase the upper limit of the degree of carbonation. In summary, a significantly accelerated carbonation effect was observed during 28 days, and thereafter, the promotional effect of UV radiation on the carbonation reaction was gradually weakened, which eventually equalised the degree of

carbonation under different conditions. Therefore, to further analyse the change in the crystallisation pattern due to the carbonation reaction under UV radiation and its effect on the microstructure of cement-based materials, the hydration products and mechanical properties of all the pastes were evaluated over 28 days of curing.

### 3.2. Hydration Phase Assemblage

### 3.2.1. Chemical Composition

In order to analyse the differences in the crystalline phases formed in the pastes cured with and without UV radiation, the samples cured for 28 days were analysed through XRD (Figure 7).

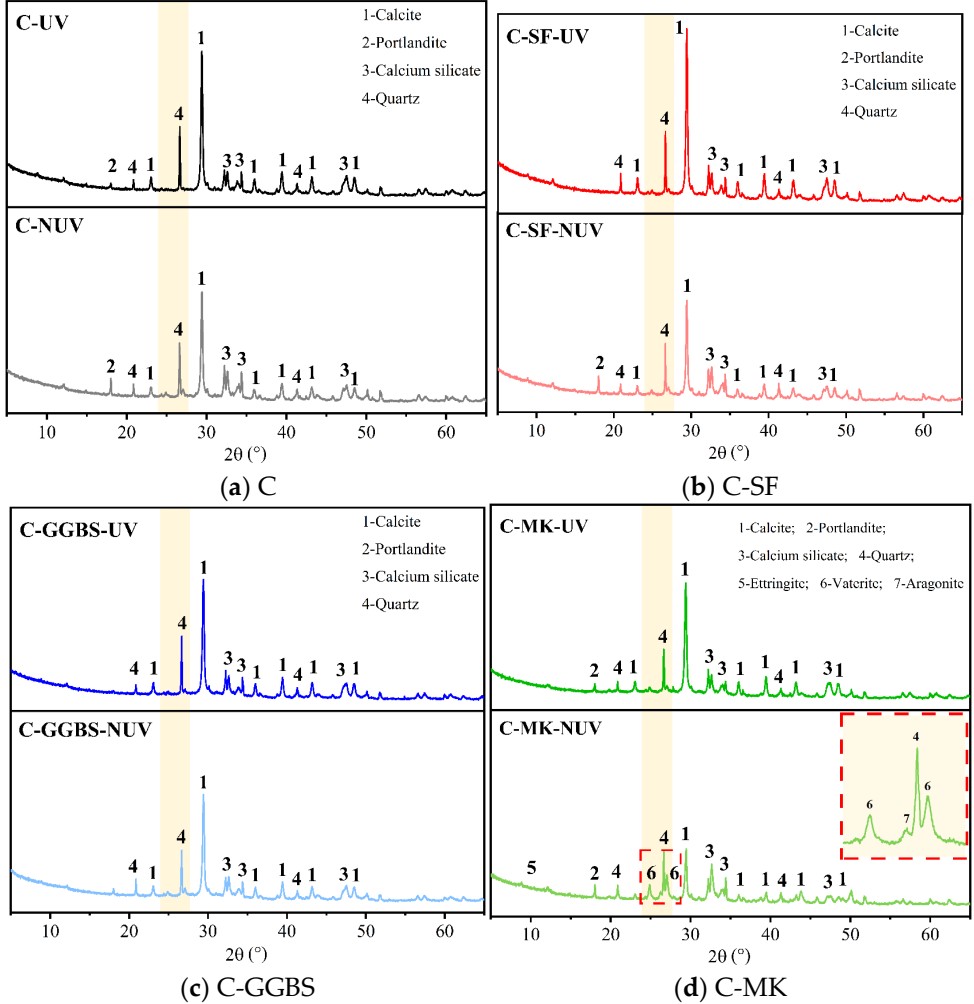

**Figure 7.** XRD patterns of samples after 28 d of curing with and without UV radiation.

As shown in Figure 7, similar changes in phase were noted in the four groups of samples. In all cases, the calcite peak ($2\theta = 29.4°$) of the UV group was stronger than that of the corresponding NUV group, whereas the calcium hydroxide peak ($2\theta = 18.0°$) was weaker. This result indicates that a greater quantity of calcite was generated due to the acceleration of the reaction between calcium hydroxide and $CO_2$ under UV radiation, which promoted the carbonation reaction. In addition, the diffraction peak intensities of $C_3S$ ($2\theta = 47.5°$) and $C_2S$ ($2\theta = 32.2°$ and $41.3°$) were also weakened to a certain extent, indicating that calcium silicate participated to a greater extent in the carbonation reaction under UV radiation. The XRD patterns of the C-SF samples (UV and NUV groups) are shown in Figure 7b. Compared with those of the NUV group, the peak corresponding to calcium hydroxide almost completely disappeared in the XRD pattern of the UV group,

whereas the peak associated with calcite was significantly intensified [31]. This result indicates that UV radiation significantly promoted the carbonation reaction and accelerated the reaction rate of calcium hydroxide, leading to well-crystallised calcium carbonate grains, which covered the surface of the UV-cured specimen. Furthermore, as shown in Figure 7c, the C-GGBS samples also exhibit a decreasing trend of the calcium hydroxide peaks and enhanced calcite peaks, but the differences between the UV and NUV groups were smaller. Compared with the other two mixtures, the NUV sample of the C-GGBS group exhibited a smaller peak intensity at $2\theta = 18°$. This result indicates that, in C-GGBS, most of the calcium hydroxide phase was consumed under non-UV conditions while further carbonation reactions of other mineral phases with low carbonation reactivity continued to occur [32,33]. Because the carbonation activity of GGBS is lower than those of $C_2S$ and $C_3S$, and both lower than that of portlandite, the pro-carbonation effect of UV radiation in the C-GGBS paste is smaller. Thus, the carbonating effect of UV radiation is difficult to distinguish through XRD analysis because of the subtle differences in the characteristic diffraction peaks [34,35]. A comparison of the patterns of the C-MK samples (Figure 7d) revealed notable crystallographic differences between the UV and NUV groups. Carbonation production in cement-based materials is commonly believed to be influenced by various factors such as their composition, pH of the pore solution, and curing conditions. Comparisons revealed that the characteristic diffraction peaks of vaterite ($2\theta = 24.9°$) and aragonite ($2\theta = 45.8°$) could be clearly identified in the XRD pattern of the NUV sample; this is because the sulphate aluminate generated in the C-MK slurry is prone to carbonation in the non-UV environment to generate vaterite and aragonite, which are usually transformed into calcite after sufficient curing time [36]. In contrast, in the UV-irradiated samples, the diffraction peaks of these two crystals could not be detected, and the calcite peaks were significantly enhanced. These results indicate that UV radiation changed the crystallisation type of the carbonation reaction or accelerated the transformation of vaterite and aragonite phases to calcite [37].

FTIR spectroscopic analysis was performed on the samples cured for 28 days under UV and non-UV conditions (Figure 8). As shown in Figure 8, all the samples provided clear FTIR bands at 713, 875, and 1427 $cm^{-1}$, confirming that calcite was the main polymorph of calcium carbonate [38]. In contrast, the FTIR peaks of vaterite and aragonite phases could not be clearly identified. This is because the FTIR bands of vaterite (at 876 $cm^{-1}$), aragonite (at 856 $cm^{-1}$), and amorphous calcium carbonate (at 866 $cm^{-1}$) overlap with those of calcite, and their contents are much lower than that of calcite [39]. Moreover, in combination with the spectra of raw materials (Figure 2), it can be found that the broad band between 900 and 1200 $cm^{-1}$ may be related to Si-O vibrations. The specimens cured with UV radiation exhibited a more pronounced signal in this range, which could be related to decalcification due to the carbonation reaction and dehydration of phases induced by UV radiation [40].

### 3.2.2. Phase Content

The degree of carbonation of the sample surface was quantitatively analysed using TGA, where DTG-DSC curves were shown in Figure 9, and the TG curves were shown in Figure 10. According to the decomposition temperature ranges of different products, the DTG curves can be divided into four periods: period I (100–400 °C), period II (400–520 °C), period III (520–720 °C), and period IV (720–950 °C). First, the weight losses of the samples during periods I and II are related to the decomposition of the hydration products, including calcium–silicate–hydrate (C-S-H), ettringite (AFt), monosulfoaluminate (AFm), and portlandite, and the weight losses during periods III and IV are due to the decomposition of carbonate phases with different crystallinities. In addition, based on the derivative TG curve, the decomposition temperature intervals of the different crystalline phases can be more accurately divided, and they can be used to characterise the phase content during each period, as shown in Figure 11. It should be noted that the weight loss ratio from TG analysis is not completely consistent with the XRD results because certain amorphous $CaCO_3$ phases in the calcium carbonate crystals cannot be detected by XRD [41,42].

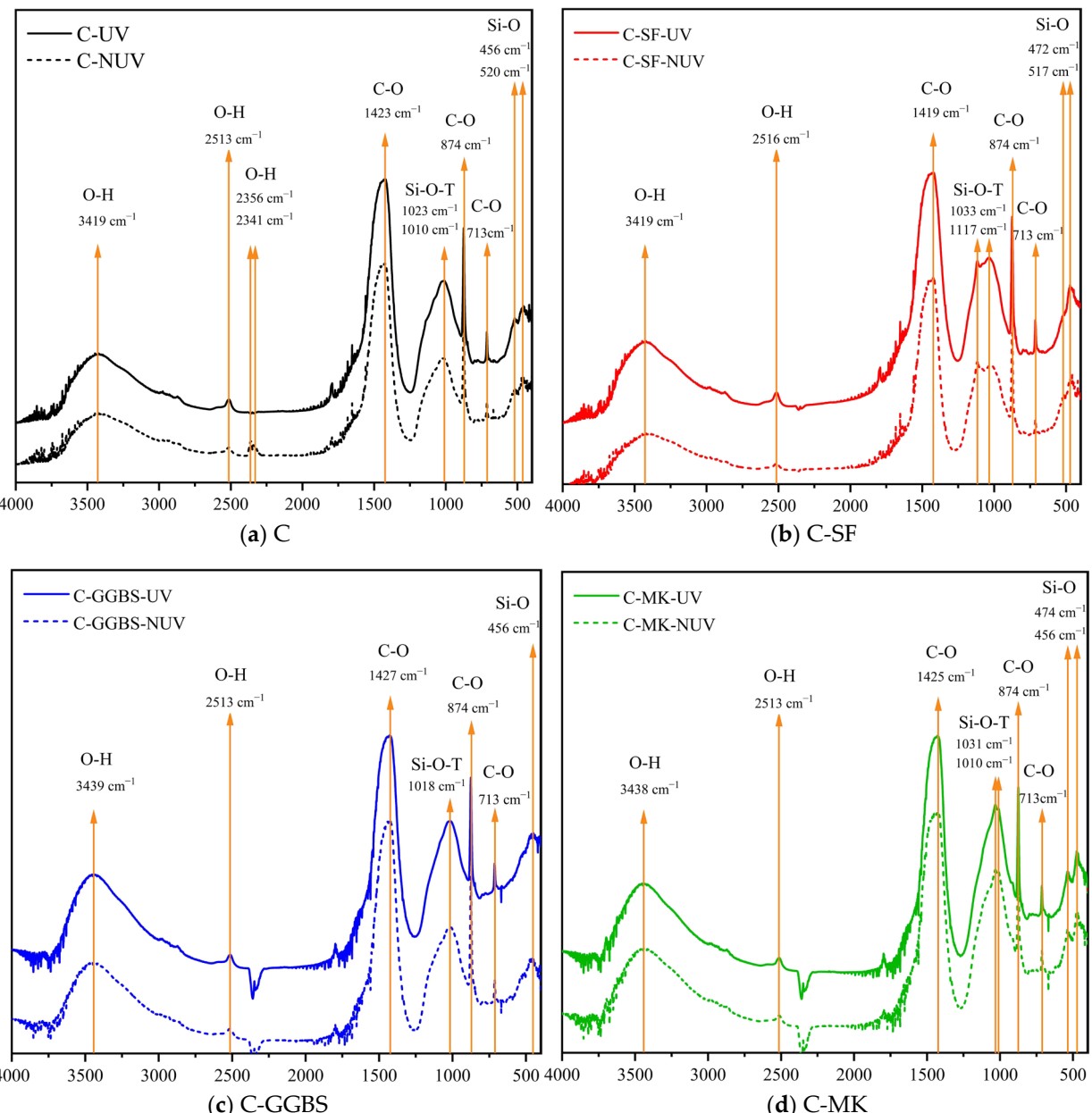

**Figure 8.** FTIR spectra of the cement samples cured for 28 days with or without UV radiation.

A larger mass loss was observed for all UV-exposed samples compared with NUV samples due to carbonate decomposition (see Figure 11); this result suggests that UV curing causes greater carbonation of the cement material. Among the four groups, the difference was most pronounced in the C-SF group. The DTG curve of the C-SF-UV specimen exhibited no notable peak between 450 and 500 °C, indicating that almost all portlandite was consumed by carbonation under the promotional effect of UV radiation. Moreover, the weight losses during periods III and IV were significantly steeper. According to the reference, the weight loss of period III is related to amorphous or poorly crystallised calcium carbonate, whereas the stable calcium carbonate phase decomposes over the higher temperature range, which is period IV. As shown in Figure 10, after 28 days of curing, the C-SF sample had the lowest calcium carbonate content among the investigated cement samples cured without UV radiation. However, under UV radiation, the weight loss in periods III and IV increased by 1.38 and 1.75 times, respectively, and the total calcium carbonate coverage of the surface layer increased by 53.3%. This change indicates that the addition of silica fume improves the early carbonation resistance of the hardened slurry in

the non-UV environment; this can be attributed to the structural compaction effect caused by the fineness of the silica fume phase. Therefore, it can be inferred that more of the hydration products participated in the carbonation reaction, and more crystalline and stable carbonate phases were generated under UV radiation [43].

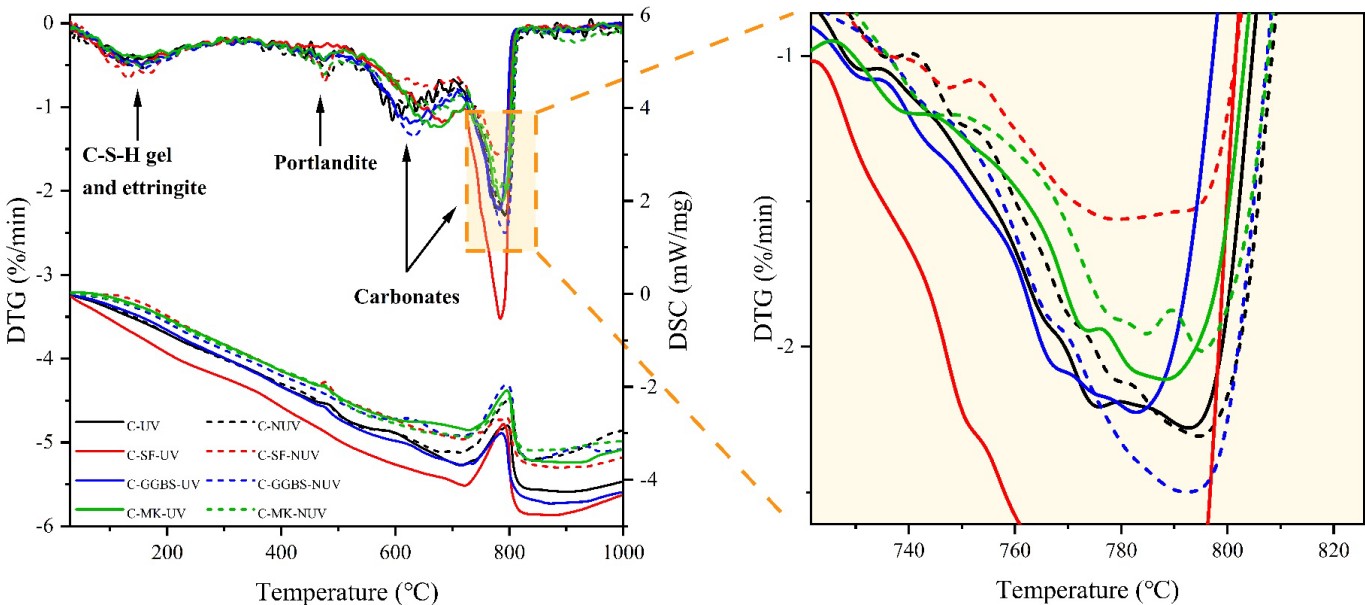

**Figure 9.** DTG-DSC curves of samples cured for 28 days with or without UV radiation.

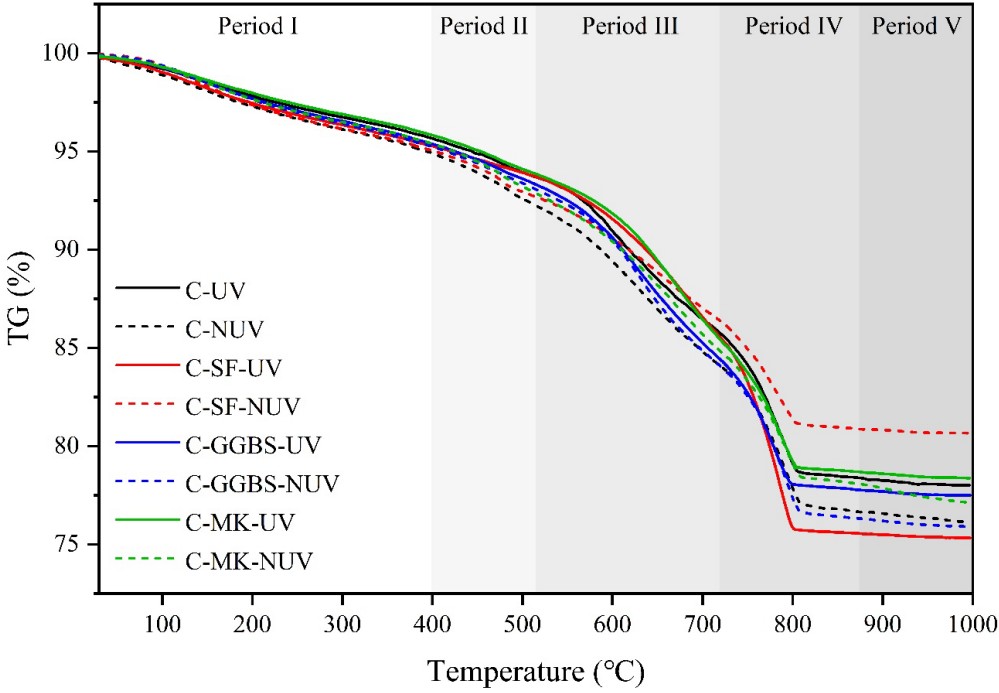

**Figure 10.** TG curves and weight loss periods of each group.

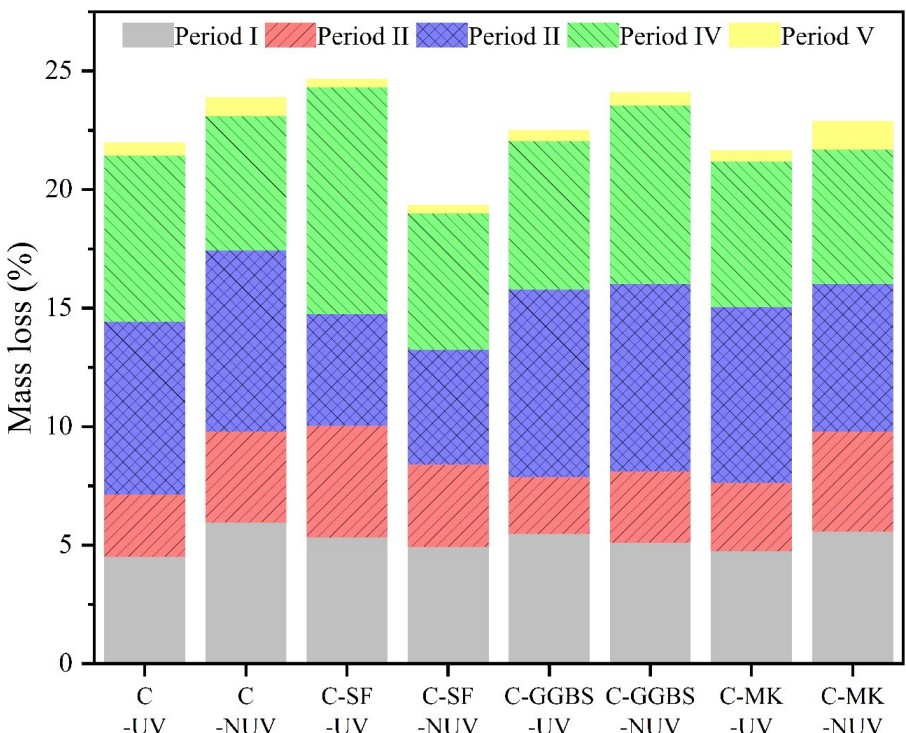

**Figure 11.** Phase content of each weight loss period in the TG curves, as calculated from the TG data.

Minor differences were observed between samples from groups C and C-GGBS; the samples from both groups contained more weakly crystallised calcium carbonate under UV radiation, which is consistent with the XRD results. For the C-MK group, a partial shoulder peak was observed at the calcium carbonate decomposition peak in period IV under the NUV condition. This can be attributed to the presence of vaterite and aragonite, and this shoulder peak disappears in the specimens under UV radiation [40].

The above analysis indicates that UV radiation can promote the carbonation reaction and facilitate the production of stable crystals in cement-based materials [44].

### 3.3. Mechanical Properties

Compressive strength and Vickers hardness were evaluated to characterise the changes in the mechanical properties of the cement samples at the macroscopic and microscopic levels, respectively. The strengths of the cement materials cured under UV and non-UV conditions were evaluated comparatively to determine the changes in the overall mechanical properties of the specimens according to the curing condition and SCM type (Figure 12). The compressive strengths of the UV groups were higher than those of the NUV groups, except for group C-MK. This result indicates that more carbonation products filled the structural pores and improved the density and mechanical properties of the cement specimens under the promotional effect of UV radiation [45,46]. In contrast, in the C-MK group, the reduction in compressive strength in the UV group is probably due to the crystalline transformation under the action of UV radiation or microcracks produced by the massive filling of carbonation products.

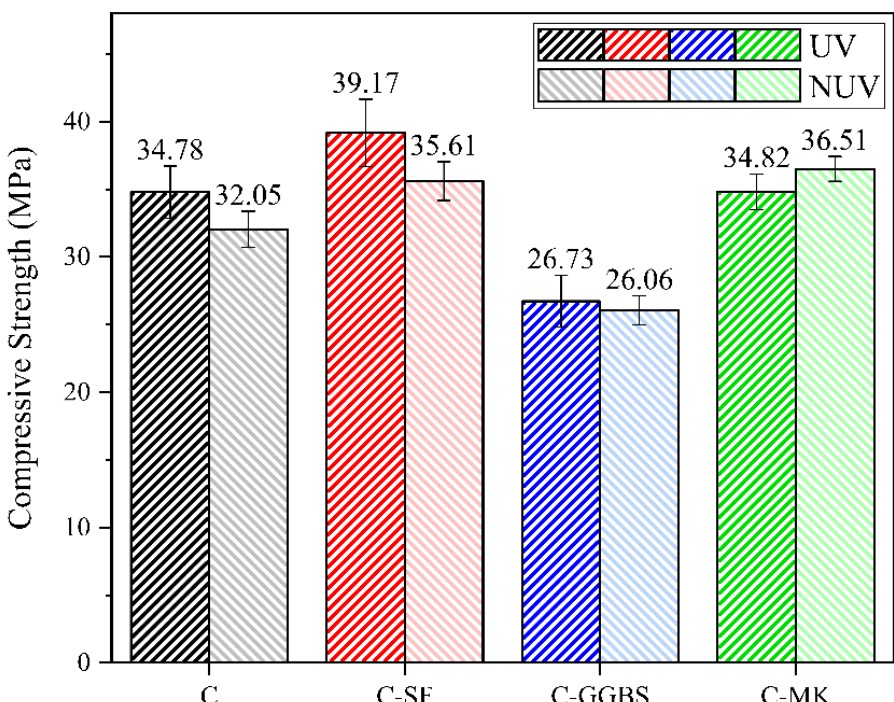

**Figure 12.** Comparison of the compressive strengths of UV and NUV samples measured on 28 days.

The Vickers hardness test was conducted using the TMVP-1 Microhardness tester, as shown in Figure 13a, and the resulting image is shown in Figure 13b. As shown in Figure 13b, the yellow area is a diamond indentation, and the red area is the calibration size. The variation pattern of the microhardness is difficult to rationalise due to the large variability in the developed composition and structure of the samples at the microscopic scale. Therefore, in this study, the surfaces of the samples were evaluated by testing the Vickers hardness values of 64 locations on an 8 × 8 grid. The data obtained were fitted by SPSS distribution, as shown in Figure 13c, and calculated to obtain its mean and standard deviation, listed in Table 3 [47]. As shown in Table 3, the samples cured under UV radiation were found to have higher mean values and standard deviation of hardness due to the UV-promoted carbonation reaction, resulting in the generation of calcium carbonate with excellent micromechanical properties in the structure composed of hydration products. This resulted in a compound enhancement in the microhardness, and it was noted that the microhardness of the samples increased with increasing degree of carbonation [48]. This result suggests that UV radiation can improve the cement microstructure and the corresponding mechanical properties in some surface areas of the hardening pastes. However, as a larger number of hydration products is involved in the carbonation reaction, a part of the original structure is destroyed, and new product phases are generated, resulting in decreased uniformity of the surface mechanical properties at the microscopic scale. Thus, the dispersion of the surface microhardness increases, resulting in a larger standard deviation of the test results.

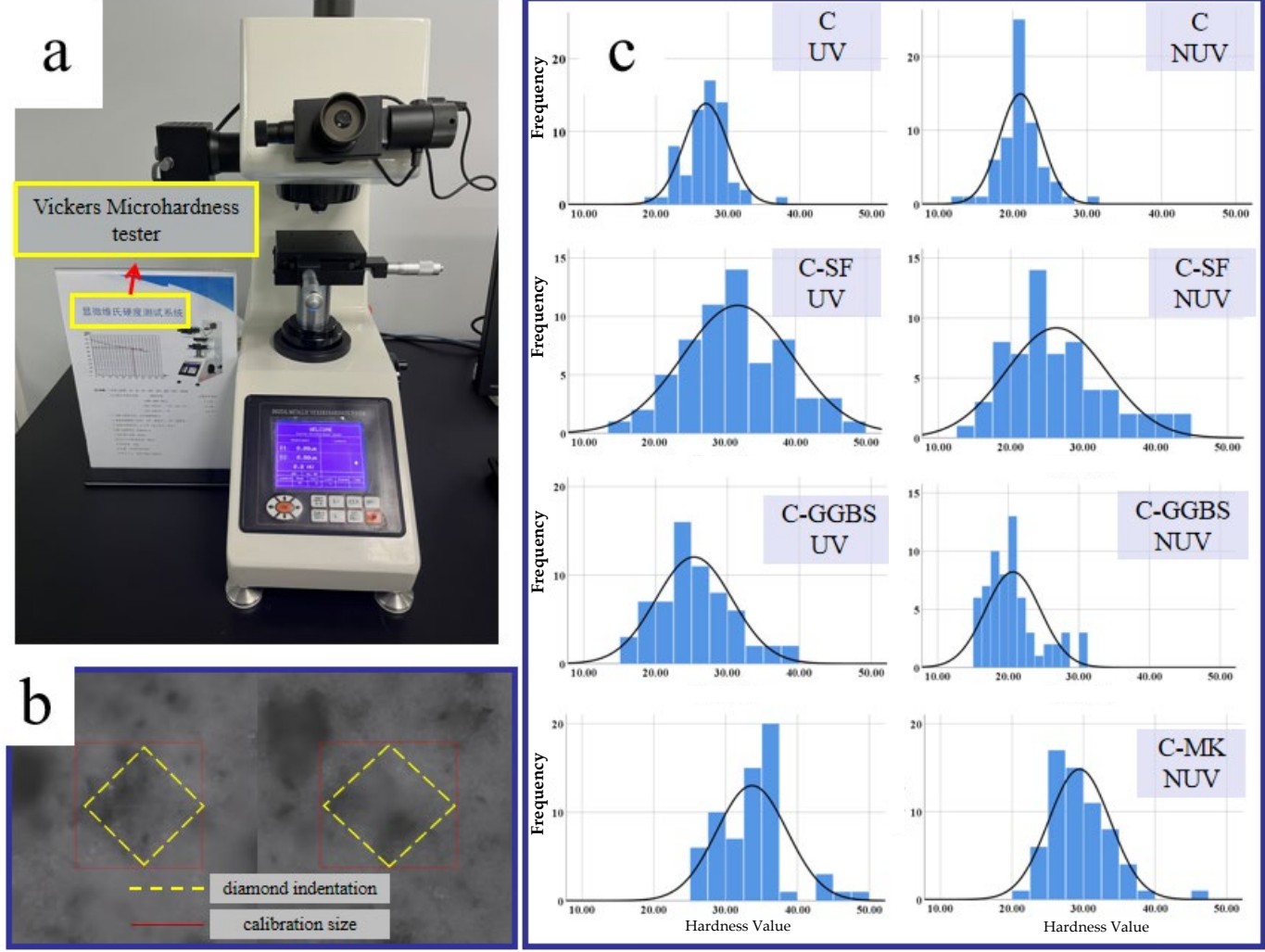

**Figure 13.** (**a**)Microhardness tester; (**b**) testing; (**c**) distribution fitting diagram.

**Table 3.** Mean and standard deviation (SD) of Vickers hardness values for different groups.

|  | C | | C-SF | | C-GGBS | | C-MK | |
|---|---|---|---|---|---|---|---|---|
|  | Mean | SD | Mean | SD | Mean | SD | Mean | SD |
| UV | 26.9 | 3.05 | 31.8 | 7.72 | 25.4 | 5.26 | 33.7 | 4.87 |
| NUV | 21.0 | 2.83 | 26.3 | 6.9 | 20.7 | 3.85 | 29.4 | 4.27 |

## 4. Conclusions

This study investigated the effects of UV radiation on the degree of carbonation, product phases, and mechanical properties of cement-based materials added with different SCMs (SF, GGBS, and MK). The following conclusions were drawn:

1.  Under UV radiation, the pH of the cement-based materials decreased significantly due to the carbonation reaction. This effect decreases with depth and increases with time in the early stage of curing, but the difference gradually decreases after 28 days of curing.

2.  The action of UV radiation on the surface of the cement-based material promoted the carbonation reaction, leading to the formation of stable calcite crystallites with excellent micromechanical properties. This structural change improved the macroscopic and micromechanical properties of the cement-based material.

3. The incorporation of SF led to a significant increase in the carbonation of the cement-based material under UV radiation, with a 53.3% increase in the amount of calcium carbonate coverage on the surface layer, whereas the addition of GGBS resulted in a lower effect of UV radiation. In practical engineering applications, the mechanical properties and carbon sequestration ability of cement-based materials can be improved by adding SF.

4. Without UV radiation, the MK-incorporated cement-based material tended to produce vaterite and aragonite phases, whereas it tended to transform into calcite under UV radiation. This transformation was accompanied by the deterioration of the macroscopic mechanical properties because UV radiation facilitated the crystallographic transformation process. Therefore, in strong UV radiation, the admixture of MK should be controlled reasonably to prevent deterioration of the macroscopic mechanical properties.

**Author Contributions:** Conceptualization, H.L. and Z.W.; methodology, H.L.; software, H.L.; validation, Z.W. and X.X.; formal analysis, H.L.; investigation, X.X.; resources, Z.S.; data curation, H.L.; writing—original draft preparation, H.L.; writing—review and editing, Z.S.; visualization, H.L.; supervision, Z.S.; project administration, Z.W.; funding acquisition, Z.W. All authors have read and agreed to the published version of the manuscript.

**Funding:** The authors gratefully acknowledge the financial support from the CSCEC Key Laboratory of Civil Engineering Materials—industrial solid waste utilization (CSCEC-PT-002), study on design and preparation of high-performance concrete using Guangdong granite-based manufacture sand (CSCEC4B-2021-KTA-11).

**Institutional Review Board Statement:** Not applicable.

**Informed Consent Statement:** Not applicable.

**Data Availability Statement:** Not applicable.

**Conflicts of Interest:** The authors declare that they have no known competing financial interests or personal relationships that could have appeared to influence the work reported in this paper.

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
