# Peer review of "Effects of UV Radiation on the Carbonation of Cement-Based Materials with Supplementary Cementitious Materials"

_coatings, doi:10.3390/coatings13060994_

Round 1

Reviewer 1 Report

An experimental study was carried out to determine the effect of UV radiation on the carbonation of cementitious composites containing different SCMs. Although the results are interesting, a major revision is necessary to consider some points in the manuscript, as follows:

-          Abstract: Please add some quantitative results at the end of the abstract.

-          Introduction, Lines 42-43: Please use a reference for this sentence.

-          Introduction, Lines 48-50: Please add references for this sentence.

-          Section 2.1: Please add the density of cement and other powders.

-          Table 2: why constant cement replacement was considered for all SCMs? The literature reported different optimum dosages. Moreover, each SCMs has various efficiency factors to be considered in the w/c ratio. These points need to be justified by the authors.

-          Section “2.2. UV-irradiation test”: Please show the actual figure for the setup of the UV test.

-          General comment: Among SCMs considered, GGBFS has lower initial stiffness, and the hydration rate is lower than other SCMs. Did the authors consider this point in the age of their tests?

-          Page 5, lines 178-179: please refer this observation to a figure.

-          The reviewer recommends merging all paragraphs related to Fig. 4 in a specific paragraph. The same as for other figures.

-          General comment: please do not use short paragraphs throughout the manuscript.

-          Regarding GGBFS, the reviewer recommends using the following reference to justify the results:

[-] Bhojaraju, C., Mousavi, S. S., & Ouellet-Plamondon, C. M. (2023). Influence of GGBFS on corrosion resistance of cementitious composites containing graphene and graphene oxide. Cement and Concrete Composites, 135, 104836.

-          Fig. 10: finding a general justification for only 28 days of curing is not reasonable for an SCMs-contained mixture. Also, comparable results were obtained for samples of UV and NUV. It is necessary to compare the 56- or 90-days curing strength. Some SCMs need Ca(OH)2 for activation, affecting the results for NUV. Accordingly, the authors should consider these details in their justifications.

-          Please add figures of damaged samples after compressive tests.

Minor editing of English language required.

Reviewer 2 Report

In this experimental study, the effect of silica fume, GGBS, and metakaolin replacements on the UV-linked carbonation behavior has been studied. The paper is quite interesting and can be accepted after following minor revisions and questions.

1-Why were the substitution levels chosen as 10% for all pozzolans? For GGBS, this ratio is usually between 20% and 80%. For silica fume and metakaolin, a replacement ratio of 10% can be acceptable. However, this should be also supported by previous studies.

The silica fume of silica fume at a level of less than 20% can be supported by citing the following papers:

https://www.mdpi.com/1996-1944/14/2/319

https://www.mdpi.com/2071-1050/13/10/5571

The use of metakaolin at a level of less than 20% can be supported by citing the following papers:

https://www.sciencedirect.com/science/article/pii/S1359836813004101

https://dergipark.org.tr/en/pub/uujms/article/164179

https://www.mdpi.com/2075-5309/12/9/1401

2- Some images of microhardness measurement should be added.

3- Hydration stoppage is a key step in the preparation of samples in the study of cement chemistry. Did the authors apply such a method?

4- Monitoring of pore structure by CT-Scan can lead to a better understanding of UV-induced carbonation. Is there any study about it?

5- In the conclusions, the effects of the main findings of this study on concrete mix design for practical engineering applications should be briefly discussed.  

Reviewer 3 Report

Line 105: the UV-irradiation test suggests using particular UV radiation for a specified time, so where are these taken from? Are these from any study or derived from some standards? Please mention the source.

Line 154: why are the samples retrieved from different depths? The UV reaction is only happening on the surface so why there is a need to collect samples at different depths.

Figure 4 needs to be incorporated in the text more efficiently. The figures are distributed among the pages.

The authors need to prepare a flow chart explaining the steps of the process followed for determining the effects of different conditions on the concrete samples.

The author has not included any figures to show the test samples, or the equipment’s that were used to complete the testing for cement box samples.

Please see the attachment for my comments in line 63-64 (page 2). This paper needs some improvement with economic analysis. The cost, and benefit was almost not touched on in the whole paper but at the end of the day, it matters to the concrete community. So please make some elaboration with respect to the current study.

Overall the paper is well written, but there is a need to revise the content and later the formatting needs to be uniform and understandable.

Looks ok to me.

Round 2

Reviewer 1 Report

The authors appropriately improved the manuscript structure.

Minor editing of English language required.